# Feasibility of linking universal child and family healthcare and financial counselling: findings from the Australian Healthier Wealthier Families (HWF) mixed-methods study

Anna M H Price  ,[1,2,3] Natalie White,[1,2] Jade Burley,[4,5,6,7] Anna Zhu,[8] Diana Contreras-Suarez,[9] Si Wang,[10] Melissa Stone,[11] Kellie Trotter,[12] Mona Mrad,[11] Jane Caldwell,[13] Rebecca Bishop,[14] Sumayya Chota,[15] Lien Bui,[16] Debbie Sanger,[17] Rob Roles,[18] Amy Watts,[1,2] Nora Samir  ,[4,5,6] Rebekah Grace,[6,19,20] Shanti Raman,[6,21] Lynn Kemp,[6,19,20,22] Raghu Lingam,[4,5,6] Valsamma Eapen  ,[6,10] Susan Woolfenden,[5,6,23] Sharon Goldfeld  [1,2,3]

SWo and SG are joint senior authors.

For numbered affiliations see end of article.

**Correspondence to**
Dr Anna M H Price;
anna.price@mcri.edu.au

## ABSTRACT

**Objectives** 'Healthier Wealthier Families' (HWF) seeks to reduce financial hardship in the early years by embedding a referral pathway between Australia's universal child and family health (CFH) services and financial counselling. This pilot study investigated the feasibility and short-term impacts of HWF, adapted from a successful Scottish initiative.

**Methods** Setting: CFH services in five sites across two states, coinciding with the COVID-19 pandemic. Participants: Caregivers of children aged 0–5 years experiencing financial hardship (study-designed screen). Design: Mixed methods. With limited progress using a randomised trial (RCT) design in sites 1–3 (March 2020–November 2021), qualitative interviews with service providers identified implementation barriers including stigma, lack of knowledge of financial counselling, low financial literacy, research burden and pandemic disruption. This informed a simplified RCT protocol (site 4) and direct referral model (no randomisation, pre–post evaluation, site 5) (June 2021–May 2022). Intervention: financial counselling; comparator: usual care (sites 1–4). Feasibility measures: proportions of caregivers screened, enrolled, followed up and who accessed financial counselling. Impact measures: finances (quantitative) and other (qualitative) to 6 months post-enrolment.

**Results** 355/434 caregivers completed the screen (60%–100% across sites). In RCT sites (1–4), 79/365 (19%–41%) reported hardship but less than one-quarter enrolled. In site 5, n=66/69 (96%) caregivers reported hardship and 44/66 (67%) engaged with financial counselling; common issues were utility debts (73%), and obtaining entitlements (43%) or material aid/emergency relief (27%). Per family, financial counselling increased income from government entitlements by an average $A6504 annually plus $A784 from concessions, grants, brokerage and debt waivers. Caregivers described benefits (qualitative) including reduced stress, practical help, increased knowledge and empowerment.

## STRENGTHS AND LIMITATIONS OF THIS STUDY

⇒ The lessons from this pilot can inform planning for implementation and evaluation at scale, to maximise the potential benefits generated by the model.
⇒ Limitations included small sample sizes, short-term follow-up, lack of control group and lack of quantitative data on non-financial impacts.
⇒ The study prioritised families who are often excluded from research and used a pragmatic and high-quality mixed-methods approach.
⇒ The iterative approach to piloting in five sites across two Australian states, and during the COVID-19 pandemic, enabled identification of design components that contributed to feasibility.
⇒ It is anticipated that the Healthier Wealthier Families model would be more feasible in the postpandemic context, as financial supports ended, and cost-of-living pressures increased.

**Conclusions** Financial hardship screening via CFH was acceptable to caregivers, direct referral was feasible, but individual randomisation was infeasible. Larger-scale implementation will require careful, staged adaptations where CFH populations and the intervention are well matched and low burden evaluation.

**Trial registration number** ACTRN12620000154909.

## INTRODUCTION

Strengthening financial security can buffer families from early adversity and enhance the environments that help children thrive.[1] During the first 1000 days of life (conception to 2 years), the human brain develops more rapidly than at any other time, laying the foundation for ongoing health and development.[2] Financial hardship can disrupt

this foundational period.[1 2] In Australia, a high-income country with universal health, education and social services, one in six children live below the poverty line, defined as 50% of median income using the traditional government measure.[3 4] One in three children experience material deprivation, missing out on essential items such as food and healthcare due to cost.[3] Even by the time Australian children start school, developmental inequities due to adverse social conditions are evident.[5] Children living in the least advantaged suburbs have three times the developmental vulnerability of the most (19% vs 7%).[5] Early adversity translates to increased risks of poorer socioemotional functioning, school failure, chronic disease, mental illness, reduced economic opportunity and intergenerational adversity.[2 6] By adopting policies and practices that address early adverse conditions, it is estimated that Australia could reduce poor developmental outcomes in middle childhood by 50%–70%.[7]

While the mechanisms are complex, it is understood that increased household income benefits children directly through better food, stable housing and healthcare (the investment model), as well as indirectly through improving parental mental health and thus parental capacity (the family stress model).[1] Healthcare offers avenues for addressing financial hardship; yet, this remains a perennial challenge for services that are designed to treat health symptoms but not their social causes.[8–10] In high-income countries with universal welfare platforms such as Australia and the UK, government assistance is available to ease financial stress. However, low take-up of benefits is common, with an estimated 40%–80% of eligible recipients missing out.[11] A total of 1.3 million Australians were found to be missing out on benefits in 2004,[12] when this issue was last publicly addressed. Common explanations for not receiving benefits include lack of awareness, system complexity and stigma.[13]

Across many high-income countries, financial counselling services are freely available to help people overcome barriers to accessing entitlements and other financial supports, typically funded by governments. In Australia, however, services remain embedded in not-for-profit organisations and unconnected to the broader system.[14] Our systematic review of health-financial linkage models for families with children aged 0–5 years identified only four international models, with limited randomised trial-level evidence.[15] Substantial benefits were demonstrated by a Scottish referral model ('Healthier Wealthier Children'), which engaged health workers in early years services to identify and refer eligible caregivers to money advice workers. Two pre–post evaluations conducted between 2010 and 2013 reported average annual financial gains per caregiver of £1661 (n=2516) and £1919 (n=2289).[16 17] Positive spill-over effects included improved health, housing and quality of life. By 2020, 10 years after it began, it had generated 27 000 referrals and over £36 million in financial gain, and is now a requirement of Scotland's Child Poverty action plan.[18]

The current study, 'Healthier Wealthier Families' (HWF), sought to adapt the Scottish model to the Australian service context and evaluate the model via a pilot randomised controlled trial (RCT).[14] Specific aims were to test the (1) feasibility of the HWF referral model and (2) short-term impacts on household finances, caregiver health, parenting efficacy and use of financial services. The overarching goal was to inform the methods and resourcing for a definitive, multisite effectiveness trial that would test whether the HWF model could reduce financial hardship for Australian families when delivered at scale.

## METHODS AND ANALYSIS
### Setting
Australia's universal Child and Family Health (CFH) services offer free, community-based healthcare to families with children from birth to school entry. Available in all Australian jurisdictions, CFH services are delivered by nurses, often in partnership with allied health providers and working with interpreters as needed. The services are mostly clinic-based and offer broad ranging support for children's health, development and parental well-being. In some jurisdictions, CFH services offer higher intensity support (more, longer and home-based appointments) for families experiencing increased health risks and social adversity. These service streams are differentiated with the labels 'universal' and 'higher intensity' throughout.

Australian CFH services already identify and respond to adverse experiences such as poor parental mental health and family violence. While the service frameworks recognise financial hardship as a social adversity affecting child and family outcomes, there is a gap in practice in systematically identifying and responding to this need. The HWF pilot was tested with CFH services in lower socioeconomic communities in Australia's two most populous states, Victoria and New South Wales. The study was planned before the COVID-19 pandemic, but implementation and evaluation occurred during the pandemic and were deeply impacted by Australia's public health restrictions and economic responses. In brief, stay-at-home orders (lockdown) were Australia's main measure of disease control until November 2021, when vaccination levels reached 70%–80%. This required CFH and financial counselling services to shift to telehealth within weeks, and saw many CFH services seconded to other pandemic-related duties. To offset the economic fallout of lockdown, the government implemented a range of financial supplements and supports in April 2020. By September 2020, this financial response had reduced poverty and housing stress beyond prepandemic levels; however, financial hardship had rebounded a year later after financial supports were withdrawn.[4] For more information on Australia's pandemic responses, please see the detailed description in online supplemental material 1.

**Figure 1** Healthier Wealthier Families data collection phases and research designs. RCT, randomised controlled trial.

## Design

When the HWF pilot study was registered with the Australian and New Zealand Controlled Trials Registry (ACTRN12620000154909, 13 February 2020), it was designed as a feasibility RCT in one Victorian site. However, implementation and evaluation during the COVID-19 pandemic required protocol iterations developed through continuous quality improvement, which incorporated quantitative and qualitative elements. When recruitment ended on 30 June 2022, we had attempted iterations of HWF in five sites across two states. Overall, the model was infeasible in sites 1–4 and feasible in site 5. Figure 1 shows the progression and design elements of the study. Recruitment began in site 1 in March 2020, before pausing 5 weeks later in response to Australia's national lockdown (and eventually ended in 2021 without recommencing). In early-2020, partnerships were formed with sites 2 and 3, reflected by updates to the trial registry (2 March 2021) and the published protocol (http://dx.doi.org/10.1136/bmjopen-2020-044488, accepted 4 May 2021).[14] Table 1 describes the RCT methods as published, following the Consolidated Standards of Reporting Trials (CONSORT) statement.[14]

To understand barriers to feasibility encountered in 2020, a qualitative implementation evaluation commenced in February 2021 (figure 1), informed by the Consolidated Framework for Implementation Research.[19] The theoretical framework was reflexive thematic analysis, and the methods are detailed in table 2. Interviews with nine service providers in sites 2 and 3 revealed three core themes, summarised below. They are described in detail with illustrative quotes in online supplemental table 1. We include these results in the methods section to explain why we had to alter the design approach.

1. Experiences and perceptions of the screening and referral processes:
   1.1. CFH providers had differing opinions about the usefulness of financial screening and relevance to their own practice, which were related to delivery of universal or higher intensity services.
   1.2. There were issues with engagement due to the research processes.
   1.3. The COVID-19 pandemic affected service provision and levels of financial hardship.
2. Barriers to identifying financial hardship and accepting help included:
   2.1. Stigma surrounding financial hardship and accepting a referral to a financial counsellor.
   2.2. Financial counselling was not well understood by CFH providers or families.

   2.3. Clients in crisis may have other priorities.
   2.4. Financial literacy and cultural and family considerations.
3. Factors that enable engagement with families with the HWF model (classified as suggestions rather than subthemes):
   – Support service providers to understand the role of financial counselling.
   – Provide resources to families about financial hardship and financial counselling.
   – Ensure accessible and inclusive communication.
   – Financial counselling services provide client-centred care, not service led.

The above themes and findings informed iterations to the HWF protocol that were trialled with two new sites (sites 4 and 5, commencing mid-late 2021, see figure 1), described next.

## Design iterations

For site 4, the RCT protocol was simplified. Research processes such as informed consent and questionnaires were shortened, and follow-up was reduced from two caregiver surveys at 3 and 6 months to one at 6 months (see table 1).

More substantive design changes were implemented for site 5. Instead of an RCT, the model employed direct referral with a pre–post evaluation (see table 1). Researcher-led processes were replaced with short, CFH-led consent and evaluation using data collected by financial counsellors rather than surveys collected by the research team. The individual randomisation was removed so there was no control group. This meant that rather than the researcher doing the referral, it was done by the CFH provider at the time of the screening survey. This made it possible for the research team to support the CFH and financial counselling providers to build trusted, working relationships with one another. The design also reflected how the referral pathway would operate in the real-world context, that is, without a researcher intervening between screening and referral.

In January–June 2022, we conducted a second phase of the implementation evaluation to understand barriers to and enablers of feasibility in sites 4–5 (figure 1, methods in table 2). This included interviews with four CFH providers in site 4, and one CFH provider and one financial counsellor in site 5 (no financial counsellors in site 4 were interviewed because no caregivers engaged, see the Results section). The themes regarding feasibility aligned with those identified in the first phase of the implementation evaluation and summarised above, so they are

**Table 1** Methods for the Healthier Wealthier Families implementation and quantitative evaluation

| Site | 1 | 2 | 3 | 4 | 5 |
|---|---|---|---|---|---|
| Design and reporting framework | Pilot RCT (as per published protocol)[14] CONSORT | | | Pilot RCT (simplified processes) CONSORT | Pre–post evaluation STROBE |
| State (metropolitan or regional) | Victoria (metropolitan) | Victoria (regional) | New South Wales (metropolitan) | New South Wales (regional) | Victoria (metropolitan) |
| Screening period | 3 March 2020 to 3 April 2020 (1 month) | 26 August 2020 to 23 November 2021 (14 months) | 2 September 2020 to 6 April 2021 (4 months) | 18 Oct 2021 to 16 May 2022 (7 months) | 21 June 2021 to 23 May 2022 (12 months) |
| CFH service tier | Universal | Higher intensity | Universal | Universal | Higher intensity |
| CFH providers screening p/week | 1–3 | 1 | 2 | 11 | 1–2 |
| Randomisation unit | Individual | | | Individual | Not applicable |
| Sample size | 180 participants (60 per site) with an anticipated 135 (75%) providing 6-month data. As Teare et al[28] report, this sample size is sufficient for informing the critical parameters when designing a definitive RCT (ie, n=70/120 for continuous/binary outcomes).[28] | | | The target was a minimum of 60 enrolled caregivers per site (30 per arm). | The target was a minimum of 30 caregivers enrolled and referred to financial counselling. |
| Participants | Caregivers attending the participating CFH services were eligible for the study if they were the caregiver of a young child aged 0–5 years old; lived within the study's geographical and service boundaries; and reported at least one risk factor for financial hardship as screened by the CFH provider (see below). The study-designed screening items were drawn from national and local datasets describing the characteristics most likely to identify families with young children who were at risk of or experiencing poverty and deprivation,[29–31] and codesigned with CFH and financial counselling partners from Site 1 (described in the protocol).[14] For sites 4–5, they were adapted following implementation evaluation and CFH provider preferences. Caregivers were excluded if they (1) could not comprehend the recruitment invitation (eg, due to substantial cognitive disability); (2) were already enrolled and assigned a research participant ID; (3) had no mechanism for contact (telephone or email); (4) already an active client in a financial counselling service. For the RCT study designs, caregivers identified as high risk for high-impact consequences were not randomised. This group (termed 'Priority 1') was referred directly to financial counselling. | | | | |
| Financial hardship criteria | 1. In the last year, because of money pressure, caregiver missed or put off: (a) mortgage or rent repayments, (b) electricity, gas, water bills, (c) buying prescription medicines, (d) paying home or car insurance. 2. No one in household has a paid job. 3. Family does not have at least $A500 in savings for an emergency. | | | As per sites 1–3, except Item 3 replaced with: Other financial pressure. | As per sites 1–3, plus the addition of: Other financial pressure, plus description. |
| Informed consent | After being sent a welcome email with links to the secure study database, consent could be completed online by the caregiver, or by phone with the researcher. | | | | Conducted with CFH provider following eligibility, recorded on study database. |
| Intervention arm | The purpose of financial counselling (the intervention) is to help clients maximise their income, reduce debt and help with budgeting. The intervention follows usual service delivery, which is tailored to a caregiver's personal and financial needs. For non-English-speaking clients, it is usual practice to deliver the service via a multilingual counsellor, or with an independent telephone interpreter. Following allocation to the intervention, a referral with the participant's contact information was emailed to the financial counsellor intake address, and a financial counsellor assigned. The financial counsellor contacted the participant to arrange the first appointment, which included confirming contact and demographic details; ascertaining capacity to make financial decisions; discussing and prioritising financial needs; and planning for working together. Caregivers were asked to gather and send paperwork to financial counsellors or external agencies and assisted by the financial counsellor to complete forms in a timely and accurate manner. Financial counsellors often contact and negotiate with creditors on a caregiver's behalf. Each participant had as many appointments with the financial counsellor as they needed, over an unbounded period (unrelated to the study timeline). | | | | |

**Table 1** Continued

| Site | 1 | 2 | 3 | 4 | 5 |
|---|---|---|---|---|---|
| Control arm | The comparator received usual care plus money advice that was freely available on the federally funded Australian Securities and Investments Commission MoneySmart website. | | | | Not applicable |
| Randomisation procedure | A statistician prepared the randomisation schedule using block randomisation. Participants were randomly assigned to either control or intervention arm with a 1:1 allocation following a computer-generated randomisation schedule stratified by site, using permuted blocks. | | | | Not applicable |
| Blinding (masking) | Participants were asked not to disclose their randomisation status during questionnaires. The research managers and participants were aware of randomisation to enable allocation and intervention delivery. Financial counsellors were informed of intervention participants but not control participants. | | | | Not applicable |
| Primary outcome (quantitative measures): Feasibility | Number/proportion of: <br>▶ Potential participants who agreed to complete screening survey (CFH report based on service records). <br>▶ Eligible caregivers who consented to participate (CFH and study records). <br>▶ Participants who accessed, attended and completed the intervention (financial counsellor report, caregiver report). <br>▶ Caregivers who completed the follow-up questionnaire (study completion records). | | | | |
| Secondary outcomes | Please see online supplemental table 3. | | | | |
| Outcome of feasibility trial for progressing to large-scale trial | Definite Go was defined as: (1) ≥50% of eligible caregivers consenting to pilot trial; (2) ≥60% of those in the intervention arm receiving the intervention; (3) ≥60% retention of consented caregivers to 6-month follow-up questionnaire. <br>Definite Stop was defined as: (1) <30% of eligible caregivers consenting to pilot trial; (2) <30% of those in the intervention arm receiving the intervention; (3) <30% retention of consented caregivers to 6-month follow-up questionnaire. | | | | |
| Fidelity measures | Financial counsellors were invited to complete a questionnaire for each intervention caregiver, designed to match usual data collection. It included issues/goals; frequency of caregiver contact and compliance; activities and advice and known changes in financial status. | | | | |
| Data collection procedures | Researchers contacted caregivers to invite them to complete baseline and follow-up questionnaires. Financial counsellors were invited to complete a 'fidelity' questionnaire for each intervention caregiver who engaged with the service, to measure the activities and advice delivered. | | | | Financial counsellor reported demographic, intake, and follow-up data for the pilot. |
| Analysis | Feasibility, caregiver characteristics and impact measures were described using proportions for categorical data and mean (SD) for continuous data. Due to the small sample sizes, no statistical analyses (described in the protocol)[14] were conducted. | | | | |

CFH, child and family health; CONSORT, Consolidated Standards of Reporting Trials; RCT, randomised controlled trial; STROBE, Strengthening The Reporting of Observational Studies in Epidemiology.

synthesised together in online supplemental table 1. We conducted interviews with eight caregivers in site 5 who engaged with financial counselling to understand perceived impacts of the HWF model (methods described in table 2). The results are presented as main findings in the next section. Due to the low levels of enrolment in sites 1–4, no caregiver interviews were conducted.

### Patient and public involvement

Patients and service providers were involved in the design and conduct of this research. During the feasibility stage, the choice and acceptability of financial hardship items was informed by discussions with CFH providers and financial counsellors, and between CFH providers and their clients. During the implementation evaluation, the trial processes were reviewed with providers in sites 2–5 and caregivers in site 5. Once the trial is published,

findings (in academic and lay outputs) will be available on the study website (https://www.rch.org.au/ccch/Healthier_wealthier_families/).

## RESULTS
### Feasibility (aim 1)
#### Completion of the financial hardship screening survey
Table 3 summarises the feasibility measures by site. The participant flow for caregivers is presented as a CONSORT diagram in online supplemental figure 1. Across the five sites, CFH providers approached a total of n=434 caregivers. In the four sites where a small number of CFH providers did HWF screening each week, 75%–100% of caseloads were approached. In site 4, where all 11 nurses did screening each week, 20% of the caseload

**Table 2** Methods for the Healthier Wealthier Families implementation and qualitative evaluation

| | |
|---|---|
| Reporting framework | Consolidated Criteria for Reporting Qualitative Research[32] |
| Purpose | To understand barriers and enablers to feasibility in sites 2–5, and the impacts for caregivers in site 5. Service changes due to the pandemic meant it was not possible to conduct the implementation evaluation in site 1. |
| Research team and reflexivity | Four researchers (denoted (R)) conducted the implementation evaluation. |
| Personal characteristics | |
| Education and occupation | R1: Master of Public Health, project coordinator for the Healthier Wealthier Families project in Victoria.<br>R2: PhD, Postdoctoral researcher at the Centre for Community Child Health, Victoria.<br>R3: PhD, Postdoctoral fellow and project coordinator for the Healthier Wealthier Families project in NSW.<br>R4: Graduate, Research assistant for the Healthier Wealthier Families project in NSW. |
| Gender | All researchers identified as female. |
| Experience and training | R1 was trained in qualitative methods and had experience on several projects conducting qualitative interviews.<br>R2 had previously conducted and published peer reviewed qualitative research.<br>R3 had previously conducted qualitative research and was in the process of publishing their research.<br>R4 had previously conducted qualitative research working with families and children. |
| Relationship with participants | |
| Relationship established, participant knowledge of interviewer | ▶ In sites 2 and 3, R3 and R4 conducted interviews with the service providers. R3 and R4 had established relationships with the team in site 3 from regular project meetings but no previous relationship with the provider in site 2.<br>▶ In sites 4 and 5, R2 conducted interviews with the service providers; there were no existing relationships. As R1 had previous interactions they did not conduct these interviews.<br>▶ In site 5, R1 and R2 conducted the caregiver interviews; there were no existing relationships. |
| Interviewer characteristics | ▶ The involvement of R1 and R3 in the conceptualisation and implementation of the project meant there was the potential for bias related to emotional investment.<br>▶ Reflexive journals were kept by R1 and R3 and used to document reflections after their interviews/focus group. |
| Study design | |
| Theoretical framework | The methodological orientation was reflexive thematic analysis. |
| Participant selection | |
| Sampling | Purposeful, convenience samples of service provider and caregiver participants were invited due to the small number of potential participants. Caregivers were remunerated with gift cards for their time in interviews ($A40 per hour). |
| Method of approach | In sites 2 and 3: The CFH management team and providers were invited to take part in the focus group and interviews by R3 via phone call. The financial counsellor was invited to take part in the interview by R4 via phone call.<br>In sites 4 and 5: Caregiver participants were asked if they would be interested in participating in an interview at their last appointment with the financial counsellor. Those that had conversational English were approached to be interviewed. Those that did not have conversational English were approached with an interpreter in their first language. In site 4, email addresses of the service providers who participated in the project were provided by the team leader. R2 emailed to invite participation in an interview. In site 5, all service providers were invited to participate in an interview via email by R2.<br>In all sites, participants (caregivers and service providers) were asked to complete a consent form before interview. |
| Sample size | ▶ In sites 2 and 3: R3 conducted a focus group with 5 CFH providers and one interview with a CFH provider.<br>▶ In site 3: R4 conducted interviews with CFH providers and one financial counsellor.<br>▶ In sites 4 and 5: R2 conducted interviews with 6 service providers one caregiver.<br>▶ In site 5: R1 conducted interviews with seven caregivers.<br>All service providers and clients identified as female. No other demographic information was collected. |

**Table 2** Continued

| Non-participation | ▶ In site 3, 5 service providers were invited to participate and none (0) declined.<br>▶ In site 4, 10 service providers were invited to participate, 4 did not respond, 1 declined, 1 agreed but was sick on the day of the interview and was not rescheduled.<br>▶ In site 5, 10 caregivers were interested in participating; however, 2 were not interviewed. One was unavailable at their scheduled interview time. One was not engaged due to difficulties in finding an interpreter in their first language. |
|---|---|
| **Setting** | |
| Setting of data collection and recording | ▶ Data in sites 2 and 3 were collected by a focus group held face-to-face and individual interviews conducted by phone. These were recorded verbatim, transcribed by an external agency, cleaned and saved as Word document.<br>▶ Data in sites 4 and 5 were collected via phone or videoconferencing. These were electronically transcribed, cleaned and saved as Word documents. |
| Presence of non-participants | None. |
| Description of sample | Site 2 was in regional Victoria, site 3 was in metropolitan NSW, site 4 was in regional NSW, site 5 was in metropolitan Victoria (see table 1). |
| **Site data collection** | |
| Interview guide | Interview guides were developed for service provider and caregiver interviews using the Consolidated Framework for Implementation Research (CFIR)[19] and included prompts from the author/researchers. They were not piloted before use. |
| Repeat interviews | Not conducted. |
| Field notes | Some field notes were made by some of the research team after interviews and the focus group. |
| Duration | The focus group was 50 min long. Interviews with service providers and caregivers were 20–50 min long. |
| Data saturation | Due to the small number of participants in this pilot study, and who were subsequently interviewed, it is unclear whether data saturation occurred. We note that the concept of data saturation is contested in the literature.[33] |
| Transcripts returned | Transcripts were not returned to participants for comment. |
| **Data analysis** | |
| Number of data coders | The four researchers who conducted interviews and focus group also coded the data, in collaboration with the project lead who did not conduct any interviews or the focus group. |
| Description of coding tree | Data were systematically categorised into themes and codes during analysis. |
| Derivation of themes | Themes were derived from the data. R1, R2, R3 and the project lead met several times to discuss themes. Themes were then finalised by R1, R2 and the project lead. |
| Software | NVivo,[34] and Microsoft Word were used for analysis of data. |
| Participant checking | No. |

CFH, child and family health; NSW, New South Wales.

was approached. Across the five sites, a total of n=355 caregivers completed the screening survey (ranging 60%–100% of those approached). Reasons for declining screening were collected in sites 1 and 5 and comprised: no interest or need for financial assistance (n=7), too personal (n=1) and no reason provided (n=8).

### Disclosure of financial hardship (eligibility)

In the universal CFH service sites, table 3 shows that n=17/47 (36%) caregivers in site 1, 29/70 (41%) in site 3 and 13/64 (20%) in site 4 disclosed financial hardship. There was greater variation between the two sites delivering higher intensity CFH services: 20/107 (19%) in site 2 compared with 66/67 (99%) in site 5. The low proportion in site 2 was at least partly due to the COVID-19

pandemic income supplements in 2020–2021 reducing the levels of financial hardship (see CFH provider quotes in online supplemental table 1 and Australian data[4]). Across the five sites, missing data on the financial screen were minimal, with only eight missing values across the items and none missing for the hardship summary variable.

### Consent/enrolment and follow-up in the study

Across the four RCT sites (1–4), 79 caregivers disclosed financial hardship and were thus eligible (table 3, online supplemental figure 1). Of these, 58 (73%) caregivers expressed interest in the study. Three caregivers were identified as high priority for asset loss or other social risk and immediately referred (see methods in table 1); 10

**Table 3** Feasibility measures by site

| Caregiver screening, eligibility and follow-up, n (%) | Pilot randomised controlled trial | | | | Pre–post evaluation |
|---|---|---|---|---|---|
| | Original (as per protocol)[14] | | | Simplified | |
| | Site 1 | Site 2 | Site 3 | Site 4 | Site 5 |
| Site characteristics | Metro Vic Universal | Regional Vic Higher intensity | Metro NSW Universal | Regional NSW Universal | Metro Vic Higher intensity |
| Potential sample size | n/a* | 107 | 117 | 411 | 69 |
| Number of caregivers approached by CFH providers | 61 (74.6)* | 107 (100) | 117 (100) | 80 (19.5) | 69 (100) |
| Of caregivers approached, n (%) who agreed to screening | 47 (77.1) | 107 (100) | 70 (59.8) | 64 (80.0) | 67 (97.1) |
| Of caregivers approached, n (%) requested/preferred interpreter | 0 | 0 | 10 (8.6) | 0 | 15 (21.7) |
| Of caregivers screened, n (%) eligible (financial hardship) | 17 (36.2) | 20 (18.7) | 29 (41.4) | 13 (20.3) | 66 (98.5) |
| Of eligible caregivers, n (%) interested in enrolling | 14 (82.4) | 9 (45.0) | 26 (89.7) | 9 (69.2) | 64 (97.0) |
| Of eligible caregivers, n (%) who enrolled | 1 (5.9) | 4 (20.0)† | 6 (20.7)† | 2 (15.4) | 49 (77)‡ |
| Of enrolled caregivers, n (%) who provided follow-up data | 1 | 4 | 0 | 0 | 44 (90)‡ |
| Breakdown of financial hardship items, n (%) who agreed to screening | | | | | |
| Missed or put off: | | | | | |
| Mortgage or rent repayments | 6 (12.8) | 0 | 15 (21.4) | 2 (3.1) | 26 (39.4) |
| Electricity, gas, water bills | 13 (27.7) | 8 (7.5) | 15 (21.4) | 9 (14.1) | 46 (68.7) |
| Buying prescription medicines | 5 (11.4) | 5 (4.7) | 12 (17.1) | 1 (1.6) | 20 (29.9) |
| Paying home or car insurance | 9 (20.0) | 2 (1.9) | 11 (15.7) | 2 (3.1) | 19 (28.4) |
| No one in household has a paid job | 0 | 20 (18.7) | 18 (25.7) | 5 (7.8) | 50 (74.6) |
| Does not have $A500 in savings | 8 (17.0) | 10 (9.4) | 14 (20.0) | Not collected | 63 (95.5) |
| Other financial pressure | Not collected | Not collected | Not collected | 4 (6.3)§ | 43 (81.1)§ |

*Data on potential caregivers provided for 3 of the 5 weeks of screening (44 screened of 59 approached).

†Enrolments included one caregiver in site 2 and two caregivers in site 3 who were immediately referred for priority indicators (not randomly allocated, see Methods in table 1).

‡Reported by the financial counsellor instead of caregivers.

§Asking about other financial pressures commenced in mid-July 2021, after 16 caregivers had been screened in site 4 (ie, denominator is n=64) and 14 caregivers had been screened in site 5 (denominator is n=53). Other financial pressures included: affording essential items (predominately food, n=16), general financial hardship (eg, low income, business strain, n=13), fines and debt (eg, car fines, childcare and Centrelink debt, n=12), unpaid bills (eg, medical, utilities, n=3) and not specified (n=3).

CFH, Child and Family Health; Metro, metropolitan; NSW, New South Wales; Vic, Victoria.

caregivers were allocated (6 intervention and 4 control); and 45/58 (78%) were lost to follow-up. Of 13 enrolled caregivers, 5 completed a follow-up questionnaire (all from sites 1 and 2). In the direct referral model in site 5, 64/67 caregivers were referred, and the financial counsellor was able to contact 49/64 (77%). In total, 44/64 (69%) engaged with the service, which included having service provision (follow-up) data recorded and provided by the financial counsellor.

### Caregivers who accessed, attended and completed the intervention

In site 3, the two caregivers who were allocated to the control group subsequently identified as high priority and were referred to financial counselling. All six caregivers enrolled in site 3 engaged with the financial counsellor, but no follow-up data were provided. Across the other RCT sites, no other intervention or priority 1 caregivers engaged with the financial counsellor. In site 5, where there were 44 attending caregivers, all appointments were conducted by phone due to the COVID-19 pandemic restrictions. Two-thirds of caregivers used one appointment; 14% used two appointments; 16% used three appointments and 2% used four appointments. Four caregivers missed one appointment and two caregivers missed two, and all were rescheduled. The financial counsellor spent an average of 1.9 hours (SD 1.1) in meetings with attending caregivers (range 0.5–4.75 hours), and an average of 2.2 hours (SD 2.2) outside of meetings on each caregiver's case (range 0.5–11 hours). By 6 months postreferral, cases were fully closed for 32/44 (73%) caregivers and partially closed (mostly due to disengagement) for 12/44 (27%).

In site 5, the most common issues facing caregivers were utility debts (32/44, 73%), awaiting government entitlements ('Centrelink', 19/44, 43%) and needing material aid or emergency relief (12/44, 27%). Eighteen per cent of caregivers had borrowed from friends and families; other debts and loans were less frequent. While no caregivers faced bankruptcy, half the caregivers disclosed priority 1 indicators, such as imminent eviction or disconnection of utilities, or being threatened with legal action (n=17/44 (39%) for each category).

### Outcome of feasibility

Overall, the feasibility measures for sites 1–4 met definite Stop criteria for progressing to a larger scale evaluation (defined in table 1) whereas for site 5 they met definite Go criteria, noting that these were defined for a pilot RCT and not a pre–post evaluation.

### Caregiver participant characteristics

Online supplemental table 2 presents the caregiver participant characteristics, noting that numbers in the RCT sites (1–4) are small and limit interpretation. Across the five sites, approximately 60% of caregivers who enrolled were partnered, and the number of children in their care ranged 1–7. Caregivers tended to have low levels of education and low household income (noting $A50 000 per annum approximated the poverty line for a family with two adults and two children). While most enrolled caregivers were not in paid work (n=7/12 in sites 1–4 and n=41/44 in site 5), some expressed a desire for more employment (n=3/9 in sites 1–4 and n=18/44 in site 5). The enrolled caregivers in sites 1–4 were more likely to have been born in Australia and speak English at home. In contrast, of the n=44 caregivers in site 5 who engaged with financial counselling, three-quarters were born overseas and just over half spoke languages other than English at home. Just under half lived with a household member with a disability or special healthcare needs. In site 5, the small number of caregivers who were referred to financial counselling but lost to follow-up (n=5) appeared broadly similar to those who engaged.

### Short-term impacts identified by quantitative evaluation (aim 2)

#### Financial impacts

The lack of follow-up data from caregivers and financial counsellors in the RCT sites 1–4 precludes the analysis described in the published Protocol.[14] For the 44 caregivers who engaged in site 5, table 4 shows that financial counselling established gains to household income including: (1) government benefits (Centrelink) totalling $A11 007 per fortnight: an average of $A355 (SD $A447) for the n=31 caregivers specified; (2) concessions totalling $A12 700: an average of $A508 for the 25 caregivers with amounts specified; (3) grants totalling $A7330: an average of $A564 (SD $A404) for the 13 caregivers with amounts specified; (4) brokerage totalling $A1680: an average of $A129 for the 13 caregivers specified and (5) a total of $A12 800 debt waived for three caregivers. When these financial gains were averaged across the 44 caregivers, income from government entitlements increased $A6504 annually and families received payments totalling $A784 from concessions, grants, brokerage and debt waivers. Table 4 shows that other frequent activities included financial literacy (66%) and referrals to other agencies (41%), most commonly housing, employment and emergency relief (see table footnote).

#### Other (non-financial) impacts

As above, the lack of follow-up data across the RCT sites (1–4) precluded the analysis of non-financial impacts described in the published protocol[14] and online supplemental table 3. In site 5, the Matthey Generic Mood Questionnaire (MGMQ) and Personal Well-being Index (PWI) were embedded into the financial counselling intake and follow-up surveys (described in online supplemental table 3). They were subsequently added to the CFH screening from October 2021 after the first 37/69 caregivers were screened. As there were high levels of missing data on these measures, we describe them in text (not tabulated). To provide a consistent sample for analysis, they are summarised for the 49 caregivers who did intake with the

**Table 4** Activities and benefits of financial counselling in site 5

| Actions and benefits recorded in routine data | No | Not applicable | Yes | If yes, amount range where specified |
|---|---|---|---|---|
| Completion of an income and expenditure statement | 29 (65.9) | 12 (27.3) | 3 (6.8) | p/fortnight: deficits of $A103, $A200, $A240 |
| Check caregiver is receiving full entitlements | 6 (13.6) | 0 | 38 (86.4) | p/fortnight: $A0–$A1400 |
| Accessed all concessions that caregiver is eligible for | 3 (6.8) | 13 (29.6) | 28 (63.6) | $A0–$A2200 |
| Is caregiver aware of Centrepay | 8 (18.2) | 18 (40.9) | 18 (40.9) | (n/a) |
| Negotiated moratorium | 5 (11.4) | 37 (84.1) | 2 (4.6) | 2 credit cards totalling $A16 000, 1 unspecified |
| Established/negotiated payment plan/s with creditor/s | 10 (22.7) | 30 (68.2) | 4 (9.1) | p/fortnight: $A10, $A180, $A200 |
| Negotiated debt reduction or waiver/s | 3 (6.8) | 38 (86.4) | 3 (6.8) | $A300, $A500, $A12 000 |
| Considered or accessed grant/s | 14 (31.8) | 15 (34.1) | 15 (34.1) | $A0–$A1330 |
| Considered or accessed no interest loan | 24 (54.6) | 10 (22.7) | 10 (22.7) | $A0–$A5000 |
| Accessed brokerage from with agency | 26 (60.5) | 4 (9.3) | 13 (30.2) | $A100–$A265 |
| Considered or accessed superannuation | 17 (38.6) | 25 (56.8) | 2 (5.6) | $A0, $A3000 |
| Organised external referrals* | 21 (47.7) | 5 (11.4) | 18 (40.9) | (n/a) |
| Conducted financial literacy activities† | 10 (22.7) | 5 (11.4) | 29 (65.9) | (n/a) |
| Achieved | | | | |
| Avoidance of bankruptcy | 0 | 44 (100) | 0 | |
| Avoidance or curtailment of legal action | 0 | 39 (88.6) | 5 (11.4) | |
| Stabilised caregiver's housing situation | 6 (14.0) | 31 (72.1) | 6 (14.0) | |
| Avoided loss of utilities | 0 | 21 (47.7) | 23 (52.3) | |
| Avoided loss of mobile phone and internet access | 0 | 42 (97.7) | 1 (2.3) | |
| Anything else the caregiver wanted to change‡ | 10 (22.7) | 1 (2.3) | 33 (75.0) | |

Centrelink is a government service which provides support to Australians who face financial hardship. Centrepay is a free bill paying service that makes regular deductions from Centrelink payments.
Fortnight: biweekly (14 days). (n/a): not applicable.
*Top 3: housing/rent assistance (n=7); employment assistance (n=6); emergency relief services (n=3).
†Top 3: financial well-being information (n=19, noting n=8 in language other than English); government MoneySmart website (n=18); budgeting with template (n=7).
‡Top 3: Centrelink entitlements (n=11); material aid and emergency relief (both n=7).

financial counsellor (relevant for 28 caregivers at CFH screening).

► At CFH screening:
– Of 28, 11 (39%) caregivers completed the MGMQ, of which 8/11 (73%) disclosed emotional distress (vs not).
– Only 4/28 (8%) caregivers completed the PWI, mean 42.3 (SD 4.9).

► At intake with the financial counsellor:
– Of 49, 43 (88%) caregivers completed the MGMQ, of which 31/43 (72%) reported high emotional distress.
– Of 49, 38 (78%) caregivers completed the PWI; mean 41.9 (SD 9.7).

► The follow-up survey was mostly completed by the financial counsellor based on client records and not in-person with the caregiver. As such, completion rates were low:
– Of 49, 7 (14%) caregivers completed the MGMQ; none reported emotional distress.
– Of 49, 11 (22%) caregivers completed the PWI; mean 57.2 (SD 11.2).

### Short-term impacts identified by qualitative evaluation (aim 2)
#### Caregiver characteristics (site 5)

In site 5, at the final appointment, the financial counsellor invited 39 caregivers to take part in the qualitative interview on behalf of the research team. Ten caregivers agreed and eight took part including one with an interpreter. Of the two who were not interviewed, one was unavailable at their scheduled time, and one was lost due to difficulties in finding an interpreter in their first

language. Demographic characteristics were not collected due to ethical approval being only available for deidentified data sharing.

## Caregiver perspectives (site 5)

Caregivers who engaged with financial counselling were unanimous in finding the intervention beneficial. Four core themes (benefits) emerged and are presented with illustrative quotes, noting some overlap between themes.

Theme 1: Reduced stress and gratitude for emotional support and advocacy received.

Caregivers described their relationship with the financial counsellor as being empathetic and respectful. The supportive nature of this relationship helped reduce the acute sense of burden and isolation caregivers were experiencing.

My financial counsellor was very supportive. She said you are not alone, there are people out there who are experiencing the same thing. So, she had a good talk to me, and after talking to her, I was able to put myself together. [CAREGIVER 4]

Caregivers were grateful for the opportunity to share their problems with a trusted and supportive service provider who listened, and for the strong advocacy they received. They experienced unexpected relief from the sense of pressure and hopelessness they had been feeling before referral to the financial counsellor.

… it's like you're sharing your feelings with someone. So I was just … I couldn't speak to anybody about it. So, she's the one who I am open with … open myself and explain my situation. And I got a very positive response from that time, so it was easy for me after that. [CAREGIVER 13]

Theme 2: Immediate solutions: crisis management and practical changes to organisation of finances.

Caregivers described several ways in which the financial counsellor assisted them. These included acting as a buffer between the caregiver and creditors; helping caregivers manage debt and bills, access entitlements to maximise their income and prevent a debt spiral; and supporting caregivers to prioritise bill payments according to their goals and needs.

She was talking to me in a way that made me feel that she would take care of everything. 'Don't worry, I will have the payments divided into instalments' … She helped me organize myself. She showed me how to organize my bills and how to make payments. [CAREGIVER 2]

This proactive and practical support provided caregivers immediate reprieve.

Because it was a big stress, for me, because always [Utility Company] send me notice—you're late to pay, you're late to pay. I can't pay anyway, I can't do it, and I was worry. You know, they send me always

notice, notice … if they cut electricity, it's hard because winter, cold, kids. So hard … So, I was so happy when I get [help]. [CAREGIVER 8]

Theme 3: Increased knowledge and clarity.

Caregivers had increased awareness of the support and options available and that they were entitled to access. Caregivers described the personalised advice about finances provided by the financial counsellor.

That's why it was really useful … she really helped with things that we have no idea existed. Can't find information on. So it was very informative. [CAREGIVER 3]

Caregivers valued the opportunity to ask questions when they did not understand something, and have their questions answered by a trusted service provider who understood their circumstances.

What I felt when they helped me with how to make payments and how to organise myself. I told you; my parents are overseas. I thought it was like a family. My father teaching me how to make payments and how to organize myself. That's my feeling.[CAREGIVER 2]

Theme 4: Well-being, self-efficacy and empowerment

Caregivers described improvements in their confidence and outlook, including more capacity and efficacy as a parent.

But my social worker and financial counsellor has been able to uplift me to make that priority along with my family's needs and open my mind and eyes to see that everything has to be on time, not to burden myself to make things worse. I know my daughter needs my attention, but I know now I have to do this first in order to be able to look after her the right way. [CAREGIVER 4]

Caregivers also described having increased control of their personal situation, more stability and feeling less stranded and stressed.

I just can say that it is very good that if woman like me will be aware that you can raise your voice and there is help … we should have more resources to have knowledge that we can go to this way. I think it's the best way if … we have more awareness for the woman who they are like me … that there is help. No matter what situation you are in, there is help. [CAREGIVER 5]

## DISCUSSION

This Australian pilot study aimed to test the feasibility and short-term impacts of a systematic referral pathway between universal CFH and financial counselling services. While financial hardship screening by CFH providers was broadly acceptable to caregivers of young children,

a feasible referral model was established in only one of five sites. Challenges to implementation were exacerbated by the COVID-19 pandemic. Online supplemental figure 2 summarises the key elements, enabling ingredients and benefits of HWF. Feasibility was enabled when screening was viewed by CFH providers as adding value to practice; direct referral was embedded between services; research activities were minimal; strong working relationships between services were enabled by the research team; CFH providers conducted longer, home-based visits and financial hardship was highly prevalent. When the feasible model was developed, financial counselling created average gains in household income from government benefits of $A125 p/week, plus additional payments totalling $A784. This is substantial given that 80% of households earned less than $A1000 per week (which approximated the Australian government's poverty line for a family with two adults and two children). While we were unable to collect quantitative data on non-financial benefits, in qualitative interviews caregivers described benefits including reduced stress and gratitude, accessing immediate solutions to their financial issues, increased knowledge and clarity, and well-being, self-efficacy and empowerment.

Australian policies are increasingly recognising the role of healthcare in addressing early adversities[20]; however, there is a dearth of research in Australia and internationally on the implementation and benefits of asking about early adversity in healthcare.[9 10] In the three pilot sites delivering universal CFH services, the 20%–41% of caregivers disclosing financial hardship was comparable with Australian population level data (also collected in 2020–2021) showing that one in three families could not afford essential items.[3] In the feasible HWF model (site 5), two-thirds of eligible caregivers engaged with financial counselling and the financial gains and sources (eg, predominately government benefits) were similar to those in the Scottish evaluations, where engagement ranged 45%–54%.[16 17] The Scottish history offers a way forward for extending the potential of the Australian model, such as engaging other early years services (eg, antenatal or early education and care) and modifying the delivery of financial counselling through colocated or outreach services.[16]

Strengths of this study included the iterative, dynamic approach to piloting in five sites across two Australian states. Adaptation of the protocol during the COVID-19 pandemic enabled identification of design components that contributed to feasibility. Recognised by the dynamic sustainability framework,[21] this approach is integral to achieving feasible, impactful implementation—and is core to the sustainability of the Scottish programme. Our study prioritised families who are often excluded from research[22] and used a pragmatic and high quality mixed-methods approach. The qualitative implementation evaluation was critical for understanding the experiences of caregivers and providers, and together with the costs and resourcing used for the pilot, can inform the

development and implementation of financial screening via healthcare on a larger scale in Australia and other high-income countries with comparable service systems.

Limitations include the small sample sizes and limited demographic data for describing the qualitative cohorts; the short-term follow-up; lack of control group; and lack of quantitative data on non-financial impacts. In site 5, the pre–post evaluation could have provided a self-control design if pre-enrolment experiences of financial activities (eg, counselling, use of MoneySmart, legal aid) were measured. Further, the high levels of missing data on emotional well-being measures reflected the difficulty in embedding new measures into CFH and financial counselling practice. Indeed, the latter does not typically inquire about caregivers' mental health needs because it can require a therapeutic response that is outside the scope of practice. Of the available data, there were high levels of emotional distress (low well-being) at screening and intake. While it was reassuring that the available measures (MGMQ and PWI) demonstrated higher emotional well-being at follow-up than intake, the sample sizes were too small to draw conclusions. The substantive challenges encountered in developing a feasible model of HWF (sites 1–4), and collecting quantitative data on non-financial impacts to enable evaluation of a feasible model (site 5) highlight the challenges that are inherent in conducting a definitive multisite effectiveness trial of HWF. Any implementation at scale will require careful and staged adaptations that test how HWF can benefit communities in the 'post-COVID-19' landscape. This includes matching of population characteristics to the healthcare intervention, and low burden evaluation through routine data collection. However, we anticipate that the HWF model would become more feasible in the postpandemic context, especially as financial supports ended, and cost-of-living pressures increased.[23]

The intention of HWF is to use universal healthcare to screen and respond to financial hardship, to prevent financial crisis and its negative impacts on children and families. The sizeable proportion of caregivers disclosing financial hardship in the universal CFH services was commensurate with the low socioeconomic status of the participating sites. Even so, screening for financial hardship challenged many providers' perception of their model of care. The levels of financial hardship did not outweigh other negatives such as stigma, lack of knowledge of financial counselling, the burden of the research processes or the pandemic disruption. There are parallels between this study and evaluation of the roll-out of family violence screening by CFH services in Victoria, which found that CFH providers selectively screened women about family violence.[24] Nurses avoided screening all women—typically those with lower incomes who are more likely to experience family violence—because of ongoing individual and system practice barriers experienced when identifying and responding to needs.[24] Unsurprisingly, in our pilot, implementing HWF was more feasible for CFH providers already providing higher

intensity, outreach support. It was in a site where almost all families were experiencing financial hardship that the HWF model added sufficient value to providers and care-givers for use. In sites where universal CFH services are provided, a tiered care model of those screening to have psychosocial adversity being supported via an integrated child and family programme (eg, a 'one stop shop' hub) incorporating HWF model might increase acceptability and engagement with the programme.

## CONCLUSION

The challenge for the Australian health system is that interventions to address financial hardship traditionally sit 'outside' the sector. This results in substantial health system inefficiencies and increased health burdens and costs.[25] Health services alone cannot redress the negative health impacts of financial hardship. The complexity of this challenge can only be meaningfully addressed by inte-grated, multisectoral and multidisciplinary approaches. As inflation and cost of living increases, we need sustain-able and cost-effective responses to financial hardship and its effects on families in the early years. There is an argument for reorienting universal services such as CFH to address the adverse social conditions facing fami-lies and driving inequitable outcomes for children and parents.[26 27]

**Author affiliations**
[1]Centre for Community Child Health, The Royal Children's Hospital, Parkville, Victoria, Australia
[2]Population Health, Murdoch Children's Research Institute, Parkville, Victoria, Australia
[3]Department of Paediatrics, The University of Melbourne, Parkville, Victoria, Australia
[4]Sydney Children's Hospitals Network Randwick, Randwick, New South Wales, Australia
[5]Women's and Children's Health, University of New South Wales, Sydney, New South Wales, Australia
[6]BestSTART-SWS, Ingham Institute, Liverpool, New South Wales, Australia
[7]Centre of Excellence for The Digital Child, The University of Wollongong, Wollongong, New South Wales, Australia
[8]School of Economics, Marketing and Finance, RMIT University, Melbourne, Victoria, Australia
[9]Melbourne Institute: Applied Economic & Social Research, The University of Melbourne, Melbourne, Victoria, Australia
[10]Psychiatry and Mental Health/ School of Clinical Medicine, University of New South Wales, Sydney, New South Wales, Australia
[11]Uniting Vic.Tas, Epping, Victoria, Australia
[12]Hume Enhanced Maternal and Child Health, Hume City Council, Hume, Victoria, Australia
[13]Wodonga Enhanced Maternal and Child Health Service, City of Wodonga, Wodonga, Victoria, Australia
[14]Wesley Mission, Sydney, New South Wales, Australia
[15]Wesley Mission, Fairfield, New South Wales, Australia
[16]Child and Family Health Services, Fairfield, New South Wales, Australia
[17]Child and Family Health Services, Albury, New South Wales, Australia
[18]Uniting Vic.Tas, Broadmeadows, Victoria, Australia
[19]Centre for the Transformation of early Education and Child Health, Western Sydney University, Sydney, New South Wales, Australia
[20]Translational Health Research Institute, Western Sydney University, Penrith South, New South Wales, Australia
[21]Community Paediatrics, South Western Sydney Local Health District, Liverpool, New South Wales, Australia
[22]Translational Research and Social Innovation (TReSI), Western Sydney University, Penrith South, New South Wales, Australia
[23]Sydney Medical School, University of Sydney, Sydney, New South Wales, Australia

**Acknowledgements** We thank all families, children, Child and Family Health providers, and financial well-being providers who collaborated on Healthier Wealthier Families (HWF). We thank members of the Australian and International HWF Advisory Groups for their oversight and guidance, especially the team at Glasgow Centre for Population Health. We thank contributors who donated time in-kind over and above the project resources which made the pilot possible, and to the funders for their flexibility around project deliverables during the COVID-19 pandemic. The HWF pilot collaboration is led by the Centre for Community Child Health and BEST START-SW, in partnership with Maternal and Child Health Services, Child and Family Health Services, Uniting Vic.Tas, Upper Murray Family Care, Wesley Mission, the Melbourne Institute: Applied Economic & Social Research, the University of Melbourne, RMIT University, Western Sydney University, and the University of New South Wales.

**Contributors** AMHP: conceptualisation, methodology, validation, formal analysis, investigation, resources, data curation, writing—original draft preparation, writing—review and editing, visualisation, supervision, project administration, funding acquisition. NW: conceptualisation, methodology, software, validation, formal analysis, investigation, data curation, writing—review and editing, supervision, project administration. JB: methodology, validation, formal analysis, investigation, resources, data curation, writing—review and editing, project administration, funding acquisition. AZ: conceptualisation, methodology, writing—review and editing, funding acquisition. DC-S: conceptualisation, methodology, writing—review and editing. SWa: investigation, writing—review and editing, project administration. MS: methodology, validation, investigation, resources, data curation, writing—review and editing, project administration. KT: methodology, validation, investigation, resources, data curation, writing—review and editing, project administration. MM: methodology, validation, investigation, resources, writing—review and editing, project administration. JC: methodology, validation, investigation, resources, data curation, writing—review and editing, project administration. RB: methodology, investigation, resources, data curation, writing—review and editing, project administration. SC: methodology, validation, investigation, resources, data curation, writing—review and editing, project administration. LB: methodology, validation, investigation, resources, data curation, writing—review and editing, project administration. DS: methodology, validation, investigation, resources, data curation, writing—review and editing, project administration. RR: methodology, validation, investigation, resources, data curation, writing—review and editing, project administration. AW: methodology, validation, formal analysis, investigation, data curation, writing—review and editing. NS: validation, formal analysis, investigation, writing—review and editing, project administration. RG: resources, writing—review and editing. SR: methodology, resources, writing—review and editing. LK: methodology, writing—review and editing. RL: methodology, resources, writing—review and editing, funding acquisition. VE: resources, writing—review and editing, funding acquisition. SWo: conceptualisation, methodology, resources, writing—review and editing, supervision, funding acquisition. SG: conceptualisation, methodology, resources, writing—review and editing, supervision, funding acquisition. All authors made substantial contributions to the conception or design of the paper; drafted this paper or revised it critically for important intellectual content; approved the final version submitted; and agree to be accountable for all aspects of the research. AMHP is responsible for the overall content as the guarantor. The guarantor accepts full responsibility for the work and/or the conduct of the study, had access to the data, and controlled the decision to publish.

**Funding** The Healthier Wealthier Families (HWF) pilot was supported by funding from the Helen Macpherson Smith Trust (Impact Grant #9523), The Corella Fund, the Murdoch Children's Research Institute (MCRI), Health@Business and University of New South Wales (UNSW) Medicine Collaboration Seed Funds Grant, the Population Child Health Group at the UNSW, Sydney Partnership for Health, Education, Research and Enterprise (SPHERE), and the Best-START SW at the Ingham Institute. The MCRI administered the philanthropic grants and provided infrastructural support (as study sponsor) to its staff but played no role in the conduct or analysis of the trial. Research at the MCRI is supported by the Victorian Government's Operational Infrastructure Support Program. AMHP was supported by The Erdi Foundation Child Health Equity (COVID-19) Scholarship; SWo was supported by a National Health and Medical Research Council (NHMRC) Career Development Fellowship (#1158954); SG was supported by an NHMRC Practitioner Fellowship (#1155290).

**Competing interests**  None declared.

**Patient and public involvement**  Patients and/or the public were involved in the design, or conduct, or reporting, or dissemination plans of this research. Refer to the Methods section for further details.

**Patient consent for publication**  Not applicable.

**Ethics approval**  This study involves human participants and this research was approved by the Human Research Ethics Committees of The Royal Children's Hospital (HREC/57372/RCHM-2019; sites 1, 2, 4–5), and South West Sydney Local Health District (2019/ETH13455; site 3). Participants gave informed consent to participate in the study before taking part.

**Provenance and peer review**  Not commissioned; externally peer reviewed.

**Data availability statement**  Data are available on reasonable request. Data are available on reasonable request to hwf.study@mcri.edu.au (subject to ethical and legal approvals).

**ORCID iDs**
Anna M H Price http://orcid.org/0000-0002-8117-8059
Nora Samir http://orcid.org/0000-0001-6571-6622
Valsamma Eapen http://orcid.org/0000-0001-6296-8306
Sharon Goldfeld http://orcid.org/0000-0001-6520-7094

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
