## [Reviewer comments · BMJ Open]

ARTICLE DETAILS

TITLE (PROVISIONAL)	The feasibility of linking universal child and family healthcare and financial counselling: findings from the Australian Healthier Wealthier Families (HWF) mixed methods study
AUTHORS	Price, Anna; White, Natalie; Burley, Jade; Zhu, Anna; Contreras-Suarez, Diana; Wang, Si; Stone, Melissa; Trotter, Kellie; Mrad, Mona; Caldwell, Jane; Bishop, Rebecca; Chota, Sumayya; Bui, Lien; Sanger, Debbie; Roles, Rob; Watts, Amy; Samir, Nora; Grace, Rebekah; Raman, Shanti; Kemp, Lynn; Lingam, Raghu; Eapen, Valsamma; Woolfenden, Susan; Goldfeld, Sharon

VERSION 1 – REVIEW

REVIEWER	Nguyen, Chi M. Indiana University Purdue University at Indianapolis, Biostatistics and Health Data Science
REVIEW RETURNED	20-Jun-2023

GENERAL COMMENTS	My concerns are somewhat aligned with the authors' stated limitations. I only have a few comments and suggestions: 1. The randomized controlled trial (RCT) in sites 1, 2, 3 and 4 unfortunately took place during the COVID-19 pandemic (from Mar 2020 to May 2022), and certainly under this pandemic's impacts. As Table 3 showed, the percentages of people who were interested in enrolling RCT were high (total of 58 persons out of 79 eligible caregivers), but the actual enroll rates dropped dramatically, resulting in only 4 persons of the control group, and 6 persons of the intervention group. I wonder if any pandemic financial supports from the Australian government would make the financial counselling in HWF less worthwhile to eligible caregivers during the study period.2. Would the HWF become more feasible under a normal context instead of the pandemic emergency context? Particularly, what would happen if the HWF was implemented after all the pandemic financial supports phase out?3. Site 5 which used the pre-post evaluation could be considered as a self-control design instead of RCT. The number of participants and the results were very encouraging. if the trial collected certain financial measures of the participants during 6 months before enrolling in HWF program, it will make the performance comparisons of a 6-month follow-up clearer and meaningful, such as:a. If participants knew and ever tried the money advice information that was freely available on the independent and government-funded website 'MoneySmart' and served as the comparator of the HWF.b. If participants had any curtailment of legal action during 6 months prior to enrollment.
--

	c. If they had any utility/sewage/phone bills unpaid during 6 months prior to enrollment. 4. It may be nice to have some figures in the main text, such as figures showing feasibility components of the RCT and the self-control trial.
--	--

REVIEWER	Song, In Han Yonsei University, Graduate School of Social Welfare
REVIEW RETURNED	17-Sep-2023

GENERAL COMMENTS	This is a study to investigate the feasibility and short-term impacts of the Healthier Wealthier Families in Australia. This research is well-structured and well-written. Only minor thing is the necessity of using mixed methods. When using mixed methods, I think having interconnected reasons to reveal the relationship between quantitative and qualitative research is good. However, in the case of this study, it is questionable whether qualitative studies of small samples are necessary to back up the results from the previous quantitative studies. Instead, I would suggest that focusing on quantitative research and omitting qualitative research will make the research more straightforward. This is because quantitative analysis alone explains the effectiveness of the HWF program well. I would like to know the authors' opinion on this suggestion. Other than that, it is evaluated as a very well-written and high-quality paper without having to point out anything else.
---

VERSION 1 – AUTHOR RESPONSE

	Reviewer comment	Author response
	Reviewer 1: Dr. Chi M. Nguyen Indiana University Purdue University at Indianapolis	
1.	My concerns are somewhat aligned with the authors' stated limitations. I only have a few comments and suggestions: 1. The randomized controlled trial (RCT) in sites 1, 2, 3 and 4 unfortunately took place during the COVID-19 pandemic (from Mar 2020 to May 2022), and certainly under this pandemic's impacts. As Table 3 showed, the percentages of people who were interested in enrolling RCT were high (total of 58 persons out of 79 eligible caregivers), but the actual enroll rates dropped dramatically, resulting in only 4 persons of the control group, and 6 persons of the intervention group. I wonder if any pandemic financial supports from the Australian government would make the financial counselling in HWF less worthwhile to eligible caregivers during the study period.	Thank you for the considered and positive review of our paper. We agree with the Reviewer and have added the following to the Discussion to reflect this "However, we anticipate that the HWF would become more feasible in the post-pandemic context, especially as financial supports ended, and cost-of-living pressures increased." We have also added the comment to the Strengths and Limitations overview: "It is anticipated that the HWF model would be more feasible in the post-pandemic context, as financial supports ended and cost-of-living pressures increased." We note that the following is already noted in the Results: "The low proportion in Site 2 was at least partly due to the

		COVID-19 pandemic income supplements in 2020-21 reducing the levels of financial hardship (see CFH provider quotes in Supplementary Table 1 and Australian data published”
2.	2. Would the HWF become more feasible under a normal context instead of the pandemic emergency context? Particularly, what would happen if the HWF was implemented after all the pandemic financial supports phase out?	Please see response to Comment 1 above.
3.	3. Site 5 which used the pre-post evaluation could be considered as a self-control design instead of RCT. The number of participants and the results were very encouraging. if the trial collected certain financial measures of the participants during 6 months before enrolling in HWF program, it will make the performance comparisons of a 6-month follow-up clearer and meaningful, such as: a. If participants knew and ever tried the money advice information that was freely available on the independent and government-funded website ‘MoneySmart’ and served as the comparator of the HWF. b. If participants had any curtailment of legal action during 6 months prior to enrollment. c. If they had any utility/sewage/phone bills unpaid during 6 months prior to enrollment.	We agree and have added the following to the Limitations section of the Discussion: “In Site 5, the pre-post evaluation could have provided a self-control design if pre-enrolment experiences of financial activities (e.g. counselling, use of MoneySmart, legal aid) were measured” .
4.	4. It may be nice to have some figures in the main text, such as figures showing feasibility components of the RCT and the self-control trial.	As suggested, we have added Figure 2, which summarises the key elements, enabling ingredients, and benefits of the HWF model.
	Reviewer 2: Prof. In Han Song Yonsei University, Harvard University T H Chan School of Public Health	
5.	This is a study to investigate the feasibility and short-term impacts of the Healthier Wealthier Families in Australia. This research is well-structured and well-written. Only minor thing is the necessity of using mixed methods. When using mixed methods, I think having interconnected reasons to reveal the relationship between quantitative and qualitative research is good. However, in the case of this study, it is questionable whether qualitative studies of small samples are necessary to back up the results from the previous quantitative studies. Instead, I would suggest that focusing on quantitative research and omitting qualitative research will make the research more straightforward. This is because quantitative analysis alone explains the effectiveness of the HWF program well. I would like to know the authors' opinion on this suggestion. Other than that, it is evaluated as a very well-written and high-quality paper without having to point out anything else.	Thank you for the considered and positive review of our paper. We agree that separating the quantitative from the qualitative results would simplify the paper and did consider writing them up separately. However, the two types of research are inextricably linked, with the first qualitative study prompted by the first quantitative study, which informed a new quantitative design and qualitative review. We have clarified this in the Design section, as follows: “However, implementation and evaluation during the COVID-19 pandemic required protocol iterations developed through continuous quality improvement, which incorporated quantitative and qualitative elements.” Many lessons regarding what did and did not work were only identified through the qualitative elements, and these lessons are fundamental to a large scale

		implementation and evaluation of HWF – which is the next step for this project. As such, we present the elements together. We recognize that the paper is long and appreciate the time the Reviewers took to review it in full.
--	--	---